# Adenomyosis and Infertility: A Literature Review

**DOI:** 10.3390/medicina59091551

**Published:** 2023-08-26

**Authors:** George Pados, Stephan Gordts, Felice Sorrentino, Michelle Nisolle, Luigi Nappi, Angelos Daniilidis

**Affiliations:** 11st Department of Obstetrics and Gynaecology, Papageorgiou General Hospital, School of Medicine, Aristotle University of Thessaloniki, 54643 Thessaloniki, Greece; angedan@hotmail.com; 2Life Expert Centre, 3000 Leuven, Belgium; stephan.gordts@lifeexpertcentre.be; 3Department of Medical and Surgical Sciences, Institute of Obstetrics and Gynecology, University of Foggia, 71121 Foggia, Italy; luigi.nappi@unifg.it; 4Department of Obstetrics and Gynecology, Hospital de La Citadelle, University of Liege, 4000 Liege, Belgium; michelle.nisolle@citadelle.be

**Keywords:** adenomyosis, diagnosis, classification, pathogenesis, infertility, treatment

## Abstract

*Background and Objectives*: Adenomyosis (the presence of ectopic endometrial glands and stroma below the endometrial–myometrial junction) is a benign condition which is increasingly diagnosed in younger women suffering from infertility. The aim of this narrative review was to study the pathophysiology and prevalence of adenomyosis, the mechanisms causing infertility, treatment options, and reproductive outcomes in infertile women suffering from adenomyosis. *Materials and Methods*: A literature search for suitable articles published in the English language was performed using PubMed from January 1970 to July 2022. *Results*: The literature search retrieved 50 articles that met the purpose of this review and summarized the most recent findings regarding the accuracy of diagnostic methods, pathophysiology, and the prevalence of adenomyosis and optimal strategies for the treatment of infertile women with adenomyosis. *Conclusions*: Adenomyosis is a common gynecological disorder, affecting women of reproductive age. It negatively affects in vitro fertilization, pregnancy and the live birth rate, as well as increases the risk of miscarriage. With the advent of non-invasive diagnoses with MRI and TVUS, the role of adenomyosis in infertility has been better recognized. Overall, more randomized controlled trials (RCTs) are needed to provide strong data on the accuracy of diagnostic methods, the pathophysiology and the prevalence of adenomyosis, the fertility outcomes of patients and the optimal strategy for the treatment.

## 1. Introduction

Adenomyosis is a benign condition of the uterus. It is characterized by the foci of the endometrial tissue invading the myometrium at a depth of at least 2.5 mm below the basal layer of the endometrium, which is typically surrounded by hyperplastic tissue. This can lead to the enlargement of the uterus. In addition, lymphatic and vascular channels penetrate the normal myometrium [1]. The symptomatology of adenomyosis typically includes chronic pelvic pain, dysmenorrhea, heavy menstrual periods, and infertility. Additionally, adenomyosis is associated with a greater incidence of anxiety, depression, and psychosocial stress [2]. However, approximately one-third of women with adenomyosis are asymptomatic. The etiology of adenomyosis is still uncertain, with many theories proposed [1,3,4,5,6,7]. A definite diagnosis of adenomyosis is made from histological examinations after hysterectomy as standard diagnostic criteria are still lacking when using imaging techniques such as a transvaginal ultrasound scan (TVUS) or magnetic resonance imaging (MRI). Thus, the prevalence of this disease has been reported with different ranges in many studies [2]. A majority of adenomyosis cases are reported in women in the fourth and fifth decades of life, while 5–25% of cases are patients younger than 39 years [6]. Recent studies have described the co-existence of adenomyosis with other pathologies such as leiomyomata (35–55% of cases) and endometriosis (65–70% of cases) [1,8]. The most common risk factors for adenomyosis are multiparity, an age of more than 40 years, and previous cesarean section or uterine surgery. However, recently, the disease has been increasingly diagnosed in younger women suffering from infertility [9]. In addition, a significant impact of adenomyosis on assisted reproductive technology outcomes has been reported [10]. The aim of this literature review was to summarize and highlight the recent data regarding the pathophysiology and prevalence of adenomyosis, the mechanisms causing infertility, treatment options, and reproductive outcomes in infertile women suffering from adenomyosis. It is very important to understand the relationship between adenomyosis and infertility. Studies are still needed to clearly define how adenomyosis affects fertility based on solid evidence of a significant association. The data would allow for appropriate counselling in these patients and the use of specific protocols in medically assisted reproduction. Well-designed research with validated data regarding an optimal strategy for the treatment of adenomyosis is needed in order to minimize bias.

## 2. Materials and Methods

We searched PubMed for relevant full-text articles published in English between January 1970 and July 2022 using the following words/keywords: ‘‘adenomyosis’’ combined with “diagnosis”, “classification”, ‘‘pathogenesis’’, “prevalence”, “infertility”, “treatment”, and “reproductive outcome”, with a restriction to human species. Data were extracted independently by the authors, who evaluated all potentially eligible papers by reading the titles and abstracts. After initial screening, citations deemed irrelevant were excluded by title. When it was not possible to assess the eligibility of an article by reading the title and abstract alone, the authors read the full text. Manual searches of review articles and cross-references completed the search. Data presented exclusively as abstracts at national and international meetings were also excluded. This study was a literature review. Patients were not involved in setting the research question or outcome measures, designing and conducting the study, or disseminating the results. No Institutional Review Board approval was required because only published and de-identified data were analyzed. The authors did not receive any specific funding for this work.

## 3. Results 

The search yielded 360 articles, all of which were available in full text on PubMed. One-hundred-and-fifty-two of these were excluded as duplicates and twenty-four were excluded as not having been written in English. Another eighty articles were excluded as it was clear from their titles and abstracts that they did not fulfill the selection criteria. We obtained the full manuscripts of the remaining 104 articles and, following the scrutiny of these, finally identified 50 relevant studies (Figure 1). Articles with an orientation to present a general overview of aspects of pathophysiology and/or surgical treatments aiming to the general symptomatology without specific orientation and detailed descriptions to the mechanisms and treatment of infertility induced by adenomyosis have been omitted. All selected articles were only articles addressing mainly the issue of fertility in order to provide an accurate and up-to-date presentation of the subject.

All fifty studies included in the final analysis were systematic reviews, randomized controlled trials, prospective or retrospective cohort studies and original articles. To ensure the highest possible quality of evidence, the authors did not only include original articles in their review. All studies investigated the diagnosis, prevalence, pathophysiology, treatment and reproductive outcomes of women with this pathology.

### 3.1. Pathophysiology and Prevalence

The etiology of adenomyosis is still not clear as the exact underlying pathogenetic mechanisms are not completely known. Endometrial glands and stroma tissue are present in the myometrium. However, at least four theories have been proposed in the recent literature that try to explain a possible pathogenesis of the disease [1,3,4,5,6,7]. The most popular theory is based on invagination of the endometrial basalis into the myometrium [1]. This can be related to the weakening of the myometrium due to previous trauma, allowing endometrial growth into the injured mucosa and stromal invasion into the inner layer of the myometrium with glandular invasion, or it can be related to an abnormal immune phenomenon involving the local production of estrogen by adenomyotic tissue, activation of macrophages and B and T cells, and production of antibodies and stimulation of cytokines. Aromatase and estrogen enzymes are present in adenomyotic tissue leading to the local production of estrogens, which might enhance the growth and expansion of the endometriotic glands and stroma into the affected myometrium. A second theory describes a de novo origin of adenomyosis from misplaced pluripotent Müllerian remnants and it is supported by studies of the proliferative and biological properties of ectopic and eutopic endometrium that demonstrate distinct characteristics. Ectopic endometrium does not have the same response in hormonal changes. Secretory changes are limited, and cyclic properties are not similar with eutopic endometrium. Biological characteristics and changes within the myometrium and expression of growth factors and cytokines seem to be completely different in adenomyosis. All these support the theory that adenomyosis has a different origin from eutopic endometrium other than basal endometrium [1,3,4,5,6,7]. A third theory suggests that invagination of the myometrial basalis proceeds along the myometrial lymphatic system, leading to adenomyosis [1,3,4,5,6,7]. Finally, a recently proposed theory purported that adenomyosis originates from bone marrow stem cells, and it provided data supporting that bone-marrow-derived stem cells contribute to the regeneration of the endometrium. This theory suggests that bone-marrow-derived stem cells might also have a contribution to the formation of the new endometrium and also repopulation of areas of myometrium leading to local proliferation of endometrial glands and stromatic tissue [1,3,4,5,6,7].

The prevalence of the disease varies from 5% to 70%. This discrepancy is strongly related to the presence of different diagnostic classification systems that lack uniformity, as well as possible pathologist bias, but it is also related to differences in the patient populations of studies. Many papers have reported direct associations between adenomyosis and multiparity, perhaps due to the invasive nature of trophoblasts and the following invagination of the basalis layer, while others have provided data supporting higher rates of adenomyosis in women who previously underwent dilation and curettage [11,12]. On the other hand, the association between adenomyosis and previous uterine surgery is still unclear, while a higher incidence of intrinsic adenomyosis has been observed in patients with a history of previously induced abortions [13]. The diagnosis of adenomyosis appears to be more common in women between 40 and 50 years old (70–80%). Additionally, data from the recent literature have shown that the prevalence of the disease in women under 39 years old varies from 5% to 25%, while in postmenopausal woman, percentages of the disease drop down to 5–10%. [6]. Data from a recent study have shown that the diffuse type of adenomyosis is more common than the focal type, and the disease develops more often in the posterior than in the anterior wall of the uterus [14]. Recently, a new theory about the evolution and, thus, the pathogenesis of uterine adenomyosis as well as peritoneal and peripheral endometriosis has been published. Levendecker and his associates have proposed that tissue injury induced by uterine hyperperistalsis/dysperistalsis and repair (TIAR) causes adenomyosis. The TIAR hypothesis has recently been expanded to lump endometriosis and adenomyosis together as one disease, called archimetriosis [15,16,17,18]. It is evident that more than one mechanism is responsible for a cascade of changes that combine some of the above theories to explain the pathogenesis of adenomyosis.

### 3.2. Genetic and Epigenetic Alteration in Adenomyosis

Recently Konincks et al. [19] have proposed the genetic–epigenetic theory for endometriosis that can be equally applied to adenomyosis, as these two conditions share common patterns of aberrant gene expression [20,21]. These include pathways that favor increased estrogen production, decreased estrogen metabolism, estrogen receptor Beta (ER-β)-driven inflammatory process, and progesterone resistance due to decreased progesterone-receptor (PR) expression. An epithelial deficiency of the enzyme HSD17β2 can lead to aromatase overexpression in endometriotic stromal cells. The same mechanism may also be involved in adenomyotic epithelial cells, as endometriosis and adenomyosis share many molecular features. Excessive levels of local estradiol in adenomyosis may be given to estrogen excess and HSD17β2 deficiency [22]. Adenomyotic tissue appears to exhibit progesterone resistance and aberrant estrogen action regulated by ER-β with excessive production of prostaglandins that cause inflammation [23].

Cytochrome P450 (CYP) genes and catechol-O-methyltransferase (COMT) gene variants could increase the risk of an estrogen-dependent disease like adenomyosis [24].

There is an increased frequency of the C allele in the T/C and C/C genotypes of the CYP1A1 gene, A allele in the C/A and A/A genotypes of the CYP1A2 gene, and the T allele in the C/T and C/C genotypes of the CYP19 gene in patients with adenomyosis [24,25].

Moreover, COMT 158 G/A gene polymorphisms contribute to the high risk of adenomyosis [24,25,26].

Epigenetic alterations have been detected in adenomyosis. Increased expression of deoxyribonucleic acid methytransferases (DNMTs) (enzymes that catalyze the transfer of a methyl group to DNA) was found in ectopic endometrium from patients with adenomyosis compared with controls [27]. Promoter hypermethylation of PR-B was detected in women with adenomyosis, leading to progesterone resistance [28].

DNA hypomethylation and increased expression of a transcription factor, CCAAT/enhancer-binding protein β were associated with the development of adenomyosis [29].

In addition to DNA methylation, the aberrant expression and localization of class I histone deacetylases (HDACs) was also detected in women with adenomyosis. Indeed, the expression of HDAC1 and HDAC3 was increased in the eutopic and ectopic endometrium of adenomyosis patients compared to controls [28]. Furthermore, the use of an HDAC inhibitor (valproic acid) is effective in treating dysmenorrhea, hyperalgesia and myometrial infiltration in patients with adenomyosis [30]. These results suggest the involvement of histone modification in the pathogenesis of adenomyosis and confirm the opinion that adenomyosis may be an epigenetic disease like endometriosis.

### 3.3. Diagnosis and Classification

Traditionally, the standard method for accurate diagnosis of adenomyosis has been hysterectomy followed by histological examination of the endometrial invasion of the underlying myometrium [9]. The presence of adenomyosis is more common in the posterior wall, less common in the anterior wall and quite rare in the cornua or in areas close to cervical os [6]. Based on histopathological examinations, adenomyosis is classified as focal if circumscribed nodules of endometrial glands and stroma surrounded by normal myometrium are found in the specimens. Diffuse adenomyosis is characterized by endometrial glands and stroma distributed throughout the myometrium. Finally, adenomyomas are considered a subgroup of focal adenomyosis surrounded by hypertrophic myometrium [9]. Many classification systems have been proposed in recent decades [12,31,32,33,34] (Table 1).

Unfortunately, the histological criteria used for the diagnosis and staging of adenomyosis were not uniform. In addition, in many cases, there was no correlation between the extension of the disease and the severity of the clinical symptoms, and some of the studies were biased; thus, none of the proposed classification systems has been generally accepted [6,9]. Recent technological advances in imaging techniques, such as TVUS and MRI, have provided clinicians with non-invasive methods for the diagnosis of adenomyosis. Recently, MUSA (morphological uterus sonographic assessment) has been proposed as a standardized method for recognizing the typical features of adenomyosis on an ultrasound assessment. These features include asymmetrical thickening of the uterine walls, intra-myometrial cysts or/and hyperechoic islands, fan-shaped shadowing on the myometrium, myometrial echogenic sub-endometrial lines and buds, trans-lesional vascularity, and an irregular or interrupted junctional zone (JZ). These features have been recently modified by the same group, considering the presence of features like cysts, hyperechogenic islands and/or echogenic sub-endometrial line bubs as diagnostic and all other features as suspicious for adenomyosis [35].

Three-dimensional (3D) TVUS can be used for better visualization of the junctional zone with a specificity of 81% and sensitivity of 85% [36]. Features of adenomyosis on 3D TVUS include an irregular, interrupted junctional zone, a junctional zone thickness > 8 mm, and a significant difference between maximum and minimum thickness measurements of the junctional zone > 4 mm [37]. In a recent meta-analysis, two-dimensional TVUS had a sensitivity and specificity of 83.8% and 63.9%, respectively, and three-dimensional TVUS had a pooled sensitivity and specificity for all combined imaging characteristics of 88.9% and 56.0%, respectively [9]. The accuracy and sensitivity of TVUS decreases to as low as 33% when a coexisting pathology such as fibroids is present, especially when the volume of the fibroid is significantly increased. MRI has also been proven to be very accurate in diagnosing adenomyosis, although it is a more expensive method compared to TVUS. MRI findings considered diagnostic for adenomyosis include a large asymmetric uterus, an abnormal junctional zone to myometrial thickness ratio of more than 40%, and junctional zone thickening of 8 to 12 mm. Recent prospective studies have shown a sensitivity of 77% and a specificity of 89% for MRI, while it seems more reasonable to opt for MRI when other uterine abnormalities such as fibroids are also present, with a sensitivity of 67% and a specificity of 82% [6]. The combination of both techniques offers the highest sensitivity for preoperative diagnosis [37]. Over the years, different classification systems have been proposed based on MRI or TVUS findings of adenomyosis in relation to histological and clinical findings of the disease [38,39,40,41,42,43,44] (Table 2).

However, none of these classification systems have been universally accepted in the recent era [9]. It remains a challenge to correlate the specific subtypes with the severity of symptoms. One of the disadvantages of most studies is that they have used a retrospective method to correlate symptoms, imaging and histology. Most studies have failed to use a properly defined prospective methodology to compare imaging, clinical symptoms and systematic microscopic evaluation of the uterus. Many authors seem to focus on the endometrium and the associated involvement of the myometrium without really evaluating the rest of the myometrium separately. Perhaps this has led to missed isolated myometrial disease. Consequently, a disease phenotype based on imaging may not correlate adequately with clinical manifestations. On the other hand, randomized prospective studies comparing imaging findings with post-hysterectomy specimens to standardize an appropriate classification system are difficult to organize. Finally, there remains the debate about the relationship and similarities between endometriosis and adenomyosis.

### 3.4. Effect of Adenomyosis on Fertility

The exact mechanism that causes infertility in women diagnosed with adenomyosis remains elusive. One of the reasons for the difficulty in accurately predicting the negative effects of adenomyosis on fertility is perhaps its high correlation with endometriosis. Adenomyosis appears to destruct the normal architecture of the myometrium, leading to the impairment of the uterine mechanisms that are important for implantation and consequent conception. The disruption of the normal junctional zone may lead to abnormal contractility, thus negatively affecting implantation. Additionally, it is not clear enough if concurrent gynecological diseases like myomas could contribute negatively to fertility. Possible mechanisms through which adenomyosis causes impairment of implantation have been described in the recent literature, including anatomical distortion of the uterine cavity, disturbed uterine peristalsis and sperm transport, dysfunctional hyperperistalsis of the inner myometrium, increased intrauterine pressure, a disturbance in normal myocyte contractility with a subsequent loss of normal rhythmic contraction, altered sex steroid hormone pathways, increased inflammatory markers and oxidative stress, the reduced expression of implantation markers, a lack of expression of adhesion molecules, and altered function of the gene for embryonic development (the HOXA 10 gene) [14]. Different locations in the female genital tract and the negative impact of adenomyosis on the individual steps of reproduction are shown in Figure 2 [45].

Other suggested mechanisms are focused on P450 (P450arom) and mRNA expression, which seem to be present in women with adenomyosis, leading to lower clinical pregnancy rates [46]. Leukemia inhibitory factor (LIF) has been demonstrated to be dysregulated in women with adenomyosis, thus impairing implantation [47]. All these factors are hypothesized to contribute to the reduction in pregnancy rates. It has become more than obvious from the literature that adenomyosis indeed has a negative impact on fertility. Recent meta-analyses have provided data associating adenomyosis and increased risk for miscarriages, 31% in women with adenomyosis and 14.1% in non-affected women [48]. The extension and type of adenomyosis appear to be important factors that affect fertility. According to a multicenter prospective study, the presence of numerous morphological features on ultrasound worsens the reproductive outcome. Clinical pregnancy decreased from 42.7% in women with no adenomyosis to 22.9% and 13.0% in those with four and seven ultrasound diagnostic features of adenomyosis, respectively [49]. A recent cross-sectional study supported that the prevalence of adenomyosis detected de novo by a 2D-TVUS in a population of young, infertile women was 7.5%. This study had some limitations as the junctional zone of the myometrium was not properly evaluated by the 2D-TVUS [6]. Endometriosis coexists in 6% to 22% of patients with adenomyosis, and leiomyomas are concurrently observed in 35% to 55% of patients [3].

A significant number of recent studies evaluating the effects of adenomyosis in infertile women undergoing assisted reproductive technology (ART) concluded that adenomyosis has a negative impact on implantation and clinical pregnancy rates in ART cycles (Table 3) [48,50,51,52,53,54,55].

A recent systematic review and meta-analysis revealed that adenomyosis is associated with a significantly lower clinical pregnancy rate (OR 0.69; 95% CI 0.51–0.94) and higher miscarriage (OR 2.17; 95% CI 1.24–3.80) rate after ART. On the other hand, no significant difference in the live birth rate (OR 0.58; 95% CI 0.29–1.17) was found. In addition, this study presented data supporting that the type of adenomyosis (focal or diffuse) does not significantly affect the reproductive outcome of patients [52]. Other studies also confirmed that rates of implantation, clinical pregnancies per cycle, clinical pregnancies per embryo transfer, ongoing pregnancies, and live birth rates among infertile women with adenomyosis undergoing IVF were significantly reduced [53,54,55]. Interestingly, several studies also demonstrated significantly higher miscarriage rates for adenomyotic women as compared to controls. Although the exact causes for the miscarriages are still not entirely clear, these results could imply a negative impact of adenomyosis [48,50]. The answer remains unclear for which type of adenomyosis has the worst fertility outcome. There is some recent evidence, though, that suggests that the focal type of the disease might have a more significant negative effect than other forms, but more evidence is definitely needed [56].

### 3.5. Treatment and Reproductive Outcomes

Treatment options (Table 4) are highly dependent upon a woman’s age, other fertility factors, and symptomatology. The small number of existing studies with limited sample sizes make it difficult to issue clear recommendations for adenomyosis and the success of reproduction.

The primary indication for the treatment of adenomyosis is the presence of symptomatology negatively affecting a patient’s daily life [1]. Although the standard method of treatment for adenomyosis is hysterectomy, the use of conservative medical or surgical options offers relief of symptoms and maintenance of fertility of patients. Nonsteroidal anti-inflammatory drugs (NSAIDs) and the hormonal control of excessive cyclic bleeding are considered the first lines of conservative medical management. Unfortunately, none of the available medical therapies can treat symptoms of adenomyosis while still allowing patients to conceive [6]. Suppressive hormonal treatments such as the continuous use of oral contraceptive pills, high-dose progestins, the levonorgestrel-releasing intrauterine device (LNG-IUD), danazol, aromatase inhibitors, selective progesterone receptor modulators, and gonadotropin-releasing hormone agonist (GnRH-a) can temporarily improve symptoms and induce the regression of adenomyosis [6,51]. Recent data have shown that only GnRH-a treatment with add-back estrogen therapy can be beneficial for infertile women with adenomyosis because of its positive effect on endometrial implantation markers, leading to improved implantation rates. In addition, a reduction of lesion size and patient quality of life has been demonstrated to be another factor which might also improve chances of conception. The long-term preparation of the endometrium with GnRH-a therapy for 2 to 4 months, before frozen embryo transfer, in women with adenomyosis undergoing IVF is associated with significantly higher clinical pregnancy, implantation, and ongoing pregnancy rates [9,50,51,52,53]. Also, pre-treatment with the LNG-IUD for 3 months before embryo transfer has been proposed to improve the reproductive outcomes of patients undergoing in vitro fertilization with a significantly increased ongoing pregnancy rate (41.8% versus 29.5%). Unfortunately, there are no published RCTs available having evaluated the efficacy of GnRH agonist pre-treatment in patients with adenomyosis. The surgical treatment of adenomyosis-related infertility remains a highly controversial issue regarding the impact of surgery on reproductive outcomes. There is still a lack of consensus on the rationale for removing the pathology in order to improve fertility. Many methods and techniques such as electrocoagulation and adenomyomectomy, with or without myomectomy, have been described, either by laparoscopy, hysteroscopy, or laparotomy. Each method has its own advantages and risks. Crucial factors to be taken into account are the proper removal of the pathology, the degree of residual disease, and the methods for setting and reconstructing the uterine wall. Proper conservative surgery could be an alternative treatment for infertile women with adenomyosis as successful pregnancies have been reported in many cases. Conservative surgical treatment aims to balance the advantages of removing the affected area against the disadvantages of leaving a possibly defective uterine wall. Factors like the extent of excision of the myometrial defect, the reconstruction technique, postoperative infection and the surgeon’s experience are quite important. Even the use of electrodiathermy instead of a cold knife during the operation might affect the wound healing and integrity of the myometrium [57]. Pertinent risks after an operation include the development of abdominal and intrauterine adhesions, placenta accreta and uterine rupture, especially during the second and third trimester of pregnancy [57]. Therefore, establishing an optimum conservative surgical technique for adenomyosis is difficult, and several operative options (open or laparoscopic), surgical techniques (complete or partial adenomyomectomy), and modified surgical techniques (U-shaped suturing, overlapping flaps, the triple-flap method, and transverse H-incisions) have been proposed. Regarding safety and the future risk of uterine rupture, for 113 women treated by the triple-flap technique, 81.4% had normal blood flow, as demonstrated by Doppler, with a 31.4% pregnancy rate and no cases of uterine rupture [57]. In women who underwent conservative surgeries for infertility treatment, pregnancy rates ranged from 25.0% to 61.5% and the miscarriage rates ranged from 11.1% to 25.0% [14]. Another recent study analyzed data from 18 facilities worldwide. Conservative surgical treatment was performed on 2365 infertile women with adenomyosis, and the postoperative pregnancy rate varied between 17.5% and 72.7%. In total, 449 pregnancies were confirmed, and 363 (80.8%) resulted in deliveries. However, artificial reproductive technology (ART) largely contributed to this relatively high pregnancy rate [57]. A review from 2014 concluded that the complete excision of localized adenomyosis in younger women is associated with a 50% delivery rate, while in women older than 40 years old, pregnancy rates were very low after cytoreductive surgery [40]. Conservative surgical treatment for uterine adenomyosis is associated with a higher risk of spontaneous rupture in a future pregnancy. A literature review suggested that the risk of uterine rupture due to pregnancy, after the removal of a uterine adenomyosis, is >1.0% compared to 0.26% in pregnancies following a myomectomy [57]. Hysteroscopy could be an alternative for cases with cavity alterations induced by adenomyosis, and metroplasty for T-shaped uteruses could lead to higher live birth rates and reduced miscarriages [58]. There is high heterogenicity between studies, and thus, a comparison of surgical techniques is not straightforward, while safety and the avoidance of complications are of the outmost importance. Perhaps individual approaches and case-by-case decision making are the most appropriate ways.

## 4. Discussion 

The current literature on adenomyosis, infertility, and reproductive outcomes demonstrates several limitations. First, the lack of standard diagnostic criteria and a globally accepted classification system for adenomyosis can easily lead to misdiagnoses [2]. In addition, the actual impact of adenomyosis on female fertility is difficult to determine due to wide varieties in size, type, localization, and severity of the disease among individuals. The presence of a concomitant pathology, including leiomyomas and endometriosis, might drastically influence the fertility of women with adenomyosis, as well. A high prevalence of endometriosis in women with adenomyosis was observed in a majority of the studies that reported on adenomyosis and fertility, and thus, the actual association of the disease with female infertility is uncertain. Destruction of normal myometrial architecture and function, disturbed uterine peristalsis and sperm transport, local hyperestrogenism, an abnormal inflammatory response, the increased presence of free radicals, and hyper vascularization are biological mechanisms that potentially relate adenomyosis to infertility and poor reproductive outcomes. All these factors make adenomyosis patients a target group for IVF. However, most of the studies comparing in vitro fertilization (IVF)/intracytoplasmic sperm injection (ICSI) outcomes between women with and without adenomyosis presented high heterogeneity regarding the age of the participants, duration of infertility, coexistence of endometriosis and leiomyoma, protocol of IVF/ICSI, number and stage of transferred embryos, and number of IVF/ICSI cycles that were carried out. In addition, most of these studies appeared to have used varying criteria for the diagnosis of adenomyosis, and the majority did not quantify the severity of the disease. Therefore, exploring or studying the correlation between adenomyosis and fertility problems currently remains difficult. In addition, since infertile patients are scrutinized more (they visit doctors more often, and more vigilant and thorough examinations and imaging methods are performed on these patients compared to healthy women), adenomyosis signs perhaps may have been reported more in this population. The published data regarding the conservative surgical treatment of women with adenomyosis have shown that it is difficult to analyze the association between the reproductive outcome and each surgical technique, as surgical techniques and managing strategies differ between facilities [1,14].

## 5. Conclusions

All the proposed pathogenetic mechanisms of adenomyosis described in the recent literature are thought to affect the receptivity of the uterine cavity and the expression of adhesion molecules necessary for embryo implantation, resulting in reduced fertility. Although diagnoses have traditionally been made through histological analyses of uterine specimens, recent data suggest that MRI and TVUS technologies are both sensitive and specific in identifying adenomyosis. TVUS seems to be the first line of investigation for suspected adenomyosis, while MRI is of the greatest value when ultrasound findings are inconclusive or other myometrial pathologies are also present. Due to a lack of uniformity among the diagnostic criteria, none of the proposed classification systems have been universally accepted. Because of the various diagnostic classifications proposed in the recent era, the prevalence of this disease in symptomatic women ranges from 5% to 70%. Perhaps a consensus of experts in the field regarding ultrasound and MRI findings and staging of the pathology might be a step forward for uniformity in classification. A validated model which would include symptoms and sonographic predictors might be useful for future clinical practice. Adenomyosis negatively impacts reproductive outcomes in patients undergoing ART. This association appears to be less significant after patients follow a long GnRH-a protocol, which improves implantation rates. GnRH-a pre-treatment can also prove beneficial prior to engaging in natural conception attempts. The role of conservative surgeries in infertile women with adenomyosis is controversial at present, as only small serial studies have shown improved reproductive outcomes. While minimally invasive procedures and ablation techniques or uterine artery embolization seem to have a role in the treatment of symptoms in women who have completed their family, there is no clear evidence for their role on fertility outcomes. Establishing an optimum conservative surgical technique for adenomyosis is difficult, and several operative options and surgical techniques have been proposed. The surgical treatment of adenomyosis-related infertility remains a highly controversial issue regarding the impact of surgery on reproductive outcomes. Adenomyosis seems to have a negative effect on the outcome of pregnancy. If clinical pregnancy is achieved, perhaps these women should receive closer antenatal follow-up. The proper assessment and management of adenomyosis is critical to the well-being of these patients. Clinicians should also always be aware of the significant impact of adenomyosis on the overall quality of life of these women, as well as the socioeconomic consequences. Apart from infertility and the high risk of miscarriage, symptoms such as heavy menstrual bleeding and chronic pelvic pain place a burden on normal activities of daily living. Overall, more randomized controlled trials (RCTs) are needed to provide strong data on the accuracy of diagnostic methods, the pathophysiology and prevalence of adenomyosis, and the fertility outcomes of patients. Well-designed research with validated data regarding an optimal strategy for the treatment of adenomyosis is needed in order to minimize bias. Finally, there is a significant necessity for uniform diagnostic criteria. Most of the discrepancies in the current available evidence are conflicting, making the accurate comparison of data practically impossible due to the different definitions and criteria.

## Figures and Tables

**Figure 1 medicina-59-01551-f001:**
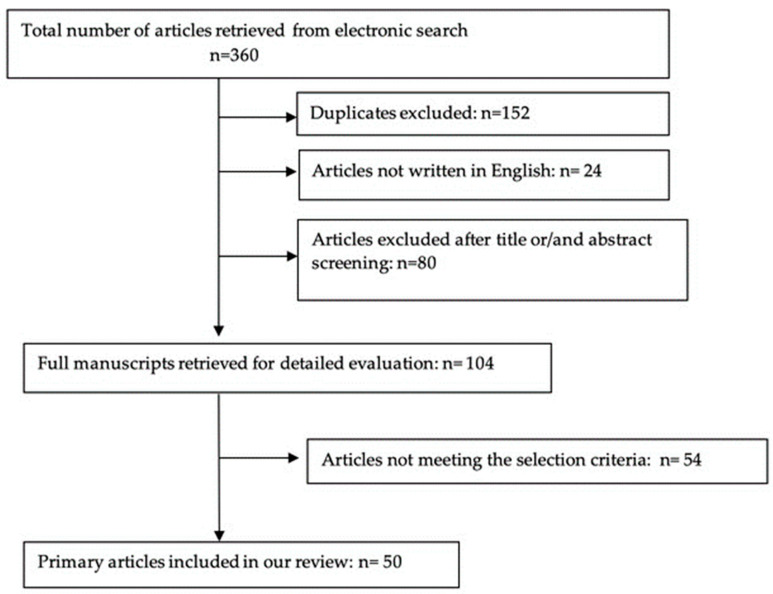
Study selection process.

**Figure 2 medicina-59-01551-f002:**
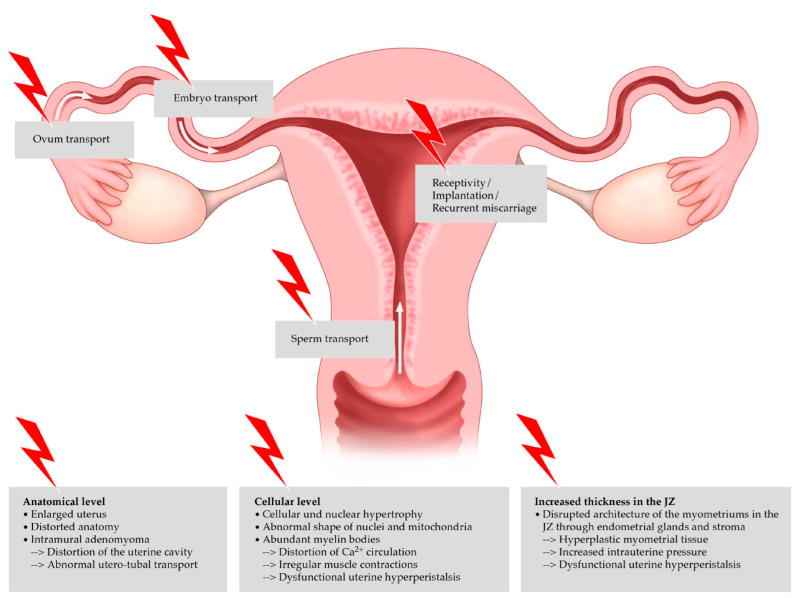
Negative impact of adenomyosis on the individual steps of reproduction.

**Table 1 medicina-59-01551-t001:** Histological classification of adenomyosis based on the depth of invasion.

Author	Year of Publication	Classification According to the Depth of Invasion
Bird et al. [31]	1972	Grade I (sub-basal lesions) Grade II (up to mid-myometrium) Grade III (beyond mid-myometrium)
Levgur et al. [12]	2000	2.5 mm depth as a cut-off from the endometrial border Superficial: <40% Intermediate: between 40–80% wall thickness Deep: >80% wall thickness
Hulka et al. [32]	2002	Category I: inner third of the myometrium Category II: focal lesions Category III: affecting the outer two-thirds of the myometrium
Sammour et al. [33]	2002	Group A: up to 25% Group B: 26–50% Group C: 51–75% Group D: >75% of myometrial thickness
Vercellini et al. [34]	2006	>2.5 mm from endometrial junction Mild: one-third of the uterine wall Moderate: two-thirds of the uterine wall Severe: more than two-thirds of the uterine wall

**Table 2 medicina-59-01551-t002:** Classification of adenomyosis based on imaging technology.

Author	Year of Publication	MRI or TVUS	Classification
Gordts et al. [38]	2008	MRI	JZ hyperplasia Adenomyosis Adenomyoma
Kishi et al. [39]	2012	MRI	Intrinsic Extrinsic Intramural All others
Grimbizis et al. [40]	2014	MRI	Diffuse Focal Polypoid Other
Bazot and Darai [41]	2018	MRI	Internal Adenomyoma External
Lazzeri L. et al. [42]	2018	TVUS	Diffuse of outer myometrium Diffuse of the inner myometrium or JZ Focal of the outer myometrium Focal of the inner myometrium Adenomyoma
Van den Bosch et al. [43]	2019		
Exacoustos et al. [44]	2020		

**Table 3 medicina-59-01551-t003:** Effects of adenomyosis on infertile women undergoing assisted reproductive technology (ART).

Author	Year	Study Design	Sample Size	Results	Limits
Vercellini et al. [48]	2014	Meta-analysis (4 prospective cohort studies and 5 retrospective cohort studies)	1865 women, 306 of which diagnosed with AD	Lower clinical pregnancy rate (PR) of 0.72 (40.5% vs. 49.8%) 2.12% higher risk of miscarriage (31.9% vs. 14.1%) Live birth rate of 0.70 (26.8% vs. 37.1%)	Qualitative and quantitative heterogeneity among studies was high
Younes and Tulandi [50]	2017	Meta-analysis (11 observational studies on clinical outcome of IVF and 4 retrospective studies evaluating the effects of surgical or medical treatment of adenomyosis on fertility)	519 patients with and 1535 without adenomyosis	Lower clinical pregnancy rate (PR) of 0.75 2.2% higher risk of miscarriage Live birth rate of 0.59	Differences in the participants’ age, duration of infertility, type of down-regulation protocol used, number and quality of the transferred embryos, number of IVF cycles performed, and the clinical outcomes assessed in the studies. In addition, the infertility diagnosis differed among studies.
Dueholm and Aagaard [51]	2018	Meta-analysis (4 case–control studies and 7 cohort studies)	1597 infertile women undergoing IVF/ICSI 782 infertile women with adenomyosis undergoing IVF/ICSI	Lower clinical pregnancy rate (PR) of 0.73 2.12% higher risk of miscarriage Live birth rate of 0.69	Only heterogeneric studies of moderate quality are available
Nirgianakis et al. [52]	2020	Meta-analysis (4 prospective studies and 13 retrospective studies)	841 women with adenomyosis undergoing ART versus 2198 women without adenomyosis undergoing ART	Lower clinical pregnancy rate (PR) of 0.69 2.17% higher risk of miscarriage No significant difference in live birth rate was found	Studies heterogeneity Diagnostic accuracy of the non-invasive imaging techniques for adenomyosis
Zhang et al. [53]	2021	Retrospective cohort study	A total of 5087 divided into two groups: adenomyosis with tubal factor infertility (study group, *n* = 193) and only tubal factor infertility (control group, *n* = 4894).	Clinical pregnancy rate 42.8% vs. 42.2% Miscarriage rate 13.3% vs. 5.6% Live birth rate 33.3% vs. 22.8%	Study design No adenomyosis classification (the severity of the disease may affect pregnancy outcomes) Diagnosis of adenomyosis by TVS is not the gold standard
Cozzolino et al. [54]	2022	Meta-analysis (7 prospective cohort studies, 15 retrospective cohort studies)	7738 patients (1277 women with adenomyosis and 6461 without adenomyosis)	Lower live birth rate (OR 0.59, 95% CI 0.37–0.92, *p* = 0.02) Lower clinical pregnancy rate (OR 0.66, 95% CI 0.48–0.90) Lower ongoing pregnancy rate (OR 0.43, 95% CI 0.21–0.88) Higher miscarriage rate (OR 2.11, 95% CI 1.33–3.33)	Studies heterogeneity (women’s age, duration of infertility, type of downregulation protocol used, number and quality of the transferred embryos, number of IVF cycles performed, and the clinical outcomes assessed in the studies) heterogeneity of the patients with different degrees of the disease (no division between focal and diffuse adenomyosis)
Liang et al. [55]	2022	Retrospective cohort study	1146 patients with adenomyosis and 1146 frequency-matched control women in a 1:1 ratio based on age, BMI, and basal follicle-stimulating hormone (FSH) level	No significant difference in clinical pregnancy rate (38.1% vs. 41.6%; *p* = 0.088) Lower implantation rate (25.6% versus 28.6%, *p* = 0.027) Lower live birth rate (26% versus 31.5%, *p* = 0.004) Higher miscarriage rate (29.1% versus 17.2%, *p* = 0.001)	Study design Diagnostic accuracy of non-invasive imaging technology for adenomyosis Inability to exclude certain pathologies, such as peritoneal endometriosis

**Table 4 medicina-59-01551-t004:** Treatment options of adenomyosis in infertility patients [6,9,45,50,51,52,53,57].

Pharmacological Treatment	Opinions/Recommendations
Nonsteroidal anti-inflammatory drugs (NSAIDs)	First-line treatment for women with pain. Negative impact on fertility.
Oral contraceptives	Treatment of pain and menstrual bleeding. No data on the impact on the subsequent fertility improvement.
GnRH analogue	Positive effect on implantation rates.
LNG-IUD	Positive effect on reproduction.
Progestins, danazol, aromatase inhibitors, selective progesterone receptor modulators	Improvement of symptoms and induction of adenomyosis. No clear data on the success of reproduction.
**Surgical treatment**	**Recommendations**
Electrocoagulation of adenomyosis foci	Positive effect on reproduction.
Adenomyomectomy with or without myomectomy	Positive effect on reproduction.

## Data Availability

All data are presented in the present manuscript.

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
