# Peer review of "Adenomyosis and Infertility: A Literature Review"

_medicina, 2023, doi:10.3390/medicina59091551_

Round 1

Reviewer 1 Report (Previous Reviewer 1)

This is a well-written and comprehensive review of the current state of knowledge on adenomyosis, with an in-depth exploration of the pathophysiology, prevalence, diagnosis, classification, impact on fertility, and treatment. The strengths and weaknesses of the review can be summarized as follows:

Strengths:

1. The extensive search strategy employed to gather relevant articles from PubMed has allowed for a broad overview of the topic.

2. The decision to include not just original articles but also systematic reviews, randomized controlled trials, and cohort studies has allowed for a comprehensive review.

3. The review successfully brings together a complex range of material on adenomyosis and presents it in a clear and concise manner.

Areas for Improvement:

1. The pathophysiology section presented four theories on the possible pathogenesis of adenomyosis. It would be useful to discuss these theories more deeply and mention if there is any consensus or predominant view in the field.

2. There is a clear need for uniform diagnostic criteria for adenomyosis, as noted in the conclusion. The authors could consider suggesting a way forward to achieve this goal.

3. The impact of adenomyosis on fertility could be expanded, specifically discussing how this disease can affect the stages of reproduction, including fertilization, implantation, and placentation. 

4. While the authors provide an overview of the current treatment options for adenomyosis, a more detailed review of these options, including surgical and medical therapies, and their effectiveness would be valuable.

5. The section on the role of imaging techniques in diagnosing adenomyosis could be updated with the latest imaging techniques and their diagnostic value.

6. In the conclusion, the authors should better synthesize the review's findings, focusing on the most critical points and explicitly stating the implications of these findings for clinical practice and future research.

7. The authors might consider providing more context about the overall impact of adenomyosis on women's quality of life, as well as a discussion of the economic burden associated with the disease.

In conclusion, this review provides a valuable contribution to the understanding of adenomyosis. By addressing the above comments, the authors can further enhance the quality and impact of their work.

The overall quality of the English language in the manuscript is very good. It is generally well-written and clear. The authors demonstrate an excellent command of scientific English, with good use of technical terminology appropriate for the topic.

However, there are a few minor areas where improvements can be made to enhance clarity and readability:

1. The phrase "Diagnoses are typically made from histologic analyses of uterine specimens; however, recent data have proven that both MRI and TVUS technologies are sensitive and specific in determining the presence of adenomyosis." could be reworded for better clarity. Consider the following: "Although diagnoses have traditionally been made through histological analyses of uterine specimens, recent data suggest that MRI and TVUS technologies are both sensitive and specific in identifying adenomyosis."

2. There are also a few instances of unnecessarily complex sentence structures that could be simplified. For example, the sentence "There is negative association between adenomyosis and reproductive outcomes in patients undergoing ART" could be simplified to "Adenomyosis negatively impacts reproductive outcomes in patients undergoing ART".

3. Careful proofreading would be beneficial to eliminate minor grammatical errors and improve clarity. 

In conclusion, with some minor revisions, this manuscript would meet the high standards expected for publication. The authors' efforts to ensure the quality of their writing are commendable.

Author Response

Reviewer 2 Report (New Reviewer)

In this review, the authors intend to summarize and highlight the recent data regarding the pathophysiology and prevalence of adenomyosis, the diagnosis and classification of adenomyosis, and causes, treatment options, and reproductive outcomes in infertile women suffering from adenomyosis. The paper has some innovation points, since the pathophysiology and classification are still not clear, and the adverse effect of adenomyosis on IVF outcomes has been widely suggested. However, due to the unspecified study design, the incompleteness of the content and the inappropriateness of the references, the results is not solid to have a reliable conclusion. My detailed comments are as follows:

1. The authors tried to cover so many aspects of adenomyosis in this mini-review, varying from pathophysiology, prevalence, diagnosis and classification to causes of infertility, treatment options and reproductive outcomes. This may make the paper appear less thoughtful, and limit the authors to discuss deeply in a specific aspect.

2. As to the methods, in Line 54, the authors indicated that the study included articles published from January 1970 to July 2021. However, the references included articles published in 2022. Besides, there is some doubt about the inclusion criteria of the review. In Line 74 and Line 77, the authors indicated that the study finally identified 32 studies, and the types varying from systematic reviews, randomized controlled studies, prospective or retrospective cohort studies, and original articles. For over 50 years, with unlimited genres, only 32 studies were chosen. Thus, there is enough reason to doubt some important relevant studies were omitted due to the narrow searching keywords. Meanwhile, it is not quite scientific proper to include studies without limitation of the types. In my opinion, a high quality review should based on previous systematic reviews with meta-analysis, or high quality original articles such as RCTs. The unselected inclusion of papers may easily resulted in omission and scientific deficiency.

3. In Line 84, the authors mentioned four theories of the etiology of adenomyosis. The context were mainly cited from references [1] and [3]. Please cited the original paper who first proposed these theories. Besides, the most popular theory should be emphasized, especially the influence factors and the newly evidence in gene mutations. The authors should included some innovative basic researches to improve scientific novelty of the current review.

4. In Line 179, the authors defined the negative impact of adenomyosis on reproduction in Figure 2, which is an original figure in reference [27]. Did the authors obtained permission from the authors and press?

5. In Line 218 and 221, Table 3 and Table 4, the tables should be combined as they were both about the IVF outcomes of women with adenomyosis. Meanwhile, the significant points of these publications should be concluded, such as sample size, grouping, type, main results and limits. And the number of included publications should be expanded, as there are so many original articles discussing the IVF outcomes of adenomyosis.

6. In Line 246 Table 5, the treatment options for adenomyosis should not be simply listing, but with opinions or recommendations.

7.  The authors should add more mostly recent publications in cited references, especially the last 3 years. 

Round 2

Reviewer 1 Report (Previous Reviewer 1)

Comments and Suggestions for Authors:

1. Title:

   - Comment: The title is descriptive and provides a clear idea about the content of the review.

   - Suggestion: Consider simplifying the title for better readability. For instance: "Adenomyosis and Infertility: A Literature Review."

2. Abstract:

   - Comment: The abstract provides an overview but lacks specific findings from the review.

   - Suggestion: Consider adopting a more structured format as per the journal guidelines. Explicitly mention methods, results, and conclusions to give readers a clearer idea of the content.

3. Introduction:

   - Comment: The introduction offers a comprehensive overview of adenomyosis.

   - Suggestion: Briefly mention the significance of understanding adenomyosis in relation to infertility to set the context for the review.

4. Diagnosis and Classification:

   - Comment: The section provides an overview of the diagnosis and classification of adenomyosis.

   - Suggestion: Discuss the challenges or limitations of current diagnostic methods to provide a more comprehensive view.

5. General:

   - Comment: There are inconsistencies in referencing and potential typographical errors.

   - Suggestion: Ensure consistent referencing throughout the manuscript. Consider having the manuscript proofread for clarity, coherence, and grammar. 

Comments on the Quality of English Language:

  1. Grammar and Syntax:

    • The manuscript generally uses correct grammar and syntax.
      1. Sentence Structure:

        • Most sentences are well-structured, but some are lengthy and could be broken down for better readability. Consider revising long sentences to make them more concise and straightforward.

Author Response

Reviewer 2 Report (New Reviewer)

The revised manuscript is well written and the overall, the methodology acceptable. Also the study question is relevant. Some minor issues should be addressed before publication.

Detailed comments,

1. In Line 347 - 352, this paragraph is not a requirement. As the authors has mentioned the argument several times.

2. According to treatment options of adenomyosis in infertility patients in Table 4, the authors  updated the table with treatment options and recommendations. One question, where are the treatment options from? Please added references in the table.

Author Response

This manuscript is a resubmission of an earlier submission. The following is a list of the peer review reports and author responses from that submission.

Round 1

Reviewer 1 Report

In this narrative review, adenomyosis was discussed from pathophysiology to infertility. The manuscript is clear, relevant to the field, and presented in a well-structured manner. The references cited are relevant, and most are recent publications. The statements and conclusions are coherent and supported by the listed citations. Overall, this exciting review adequately mentions and discusses the relevant literature. However, I have some comments or suggestions to improve the quality of the manuscript.

1.         Recently, a new theory on adenomyosis has been published. Archimetrosis, a new theory on the pathogenesis of uterine adenomyosis and endometriosis related to stratum vasculare development, tissue injury, and repair. (Leyendecker, G.; Wildt, L.; Laschke, M.W.; Mall, G. Archimetrosis: The evolution of a disease and its extant presentation: Pathogenesis and pathophysiology of archimetrosis (uterine adenomyosis and endometriosis). Arch. Gynecol. Obstet. 2022)

2.         For the benefit of the reader, a figure or table showing the adverse effects of adenomyosis on reproductive outcomes should be included.

3.         A column showing IVF outcomes in women without adenomyosis should be added to Table 3.

4.         Treatment of adenomyosis is by medication and surgery. A table would help the reader better understand the treatment options.

5.         For the benefit of the reader, reproductive outcomes after treatment for adenomyosis should be detailed.

Reviewer 2 Report

Adenomyosis is an important benign uterine lesion with great clinical implications to the patients. Hence updating the recent literature is important. However, there are many recent publications including reviews, which were not included in the discussion such as following to mention a few.

Adenomyosis: An update regarding its diagnosis and clinical features.

Bourdon M, Santulli P, Marcellin L, Maignien C, Maitrot-Mantelet L, Bordonne C, Plu Bureau G, Chapron C.J Gynecol Obstet Hum Reprod. 2021 Dec;50(10):102228. doi: 10.1016/j.jogoh.2021.102228. Epub 2021 Sep 11.

Adenomyosis and Infertility-Review of Medical and Surgical Approaches.

Szubert M, Koziróg E, Olszak O, Krygier-Kurz K, Kazmierczak J, Wilczynski J.Int J Environ Res Public Health. 2021 Jan 30;18(3):1235. doi: 10.3390/ijerph18031235.

Adenomyosis and infertility.

Moawad G, Kheil MH, Ayoubi JM, Klebanoff JS, Rahman S, Sharara FI.J Assist Reprod Genet. 2022 May;39(5):1027-1031. doi: 10.1007/s10815-022-02476-2. Epub 2022 

Uterine polyps, adenomyosis, leiomyomas, and endometrial receptivity.

Munro MG.Fertil Steril. 2019 Apr;111(4):629-640. doi: 10.1016/j.fertnstert.2019.02.008.

denomyosis pathogenesis: insights from next-generation sequencing.

Bulun SE, Yildiz S, Adli M, Wei JJ.Hum Reprod Update. 2021 Oct 18;27(6):1086-1097. doi: 10.1093/humupd/dmab017.

So it needs to be seen whether the present review is offering any new information on the subject. 

References of theories may be added again after citing each, especially the 4th one (line 97)

Line 117: ‘in case that’- this is confusing. May be removing ‘that’ will not affect what authors want to say. Or ‘is found on the specimen’ may be removed.

Table-1: There may be a distinct gap between 2 consecutive classifications, as there seem to be an overlap. Also alignment of last column may me kept as ‘aligned from left’ rather than ‘centre’. 

Lines 141-142: defines sensitivity and specificity of which ‘method’?

Table-2 and Table-3: Alignment of last column may me kept as ‘aligned from left’ rather than ‘centre’. 

Line 153: co-existing is better that coexistent

Lines163-164: ‘through------literature’ should be deleted as it is being repeated. 

Line 166: LIF should come later in bracket.

Line 168: ‘in combintion’ may be removed.

Line 169-170: ‘of women-------- investigation--------fertility.’ may be deleted.

Line 172: and may be added between ‘adenomyosis’ and ‘increased’. And full stop may be replaced wuth comma with no need for putting the bracket.

Lines 178-180 and 183-186: ‘Additional --- evaluation’ ; ‘however------recent studies [3,10]-----not clear’ may be deleted; as same thing is written again and again throughout the paper. 

Lines 188-191: Both sentences may be combined. 

Lines 194 to 209: A recent systematic review and meta-analyses revealed 

                                    In addition, this study presented data supporting that

                                    Another recent systematic review, also, confirms that

                                    several studies also demonstrated

                                    The answer remains still unclear 

                                    There is some recent evidence though which suggests that

This para contains lots of such sentences. This para may be re-written with only the crux with references in brackets, rather than these decorative lines, which is diluting the basic content.

In fact, whole paper has abundance of such sentences which can be reduced. Since the article is on ‘Adenomyosis’, authors may also omit repeating ‘of adenomyosis’ eg. In lines 212, 214, 218 and so on.

226: leading ‘to’ rather than ‘on’

Paragraphs are very long and can be broken into several paragraphs. 

Line 231: ‘There is still lack of concensus to the rationate to’ – Overuse of such sentences will make readers lose interest.  

There are many spelling errors: eg residual in line  236; repetition of risk in line 259, sutting etc.